# The Relationship between the Concentration of Magnesium and the Presence of Depressive Symptoms and Selected Metabolic Disorders among Men over 50 Years of Age

**DOI:** 10.3390/life11030196

**Published:** 2021-03-03

**Authors:** Iwona Rotter, Adrian Wiatrak, Aleksandra Rył, Katarzyna Kotfis, Olimpia Sipak-Szmigiel, Magdalena Ptak, Natalia Tomska, Aleksandra Szylińska

**Affiliations:** 1Department of Medical Rehabilitation and Clinical Physiotherapy, Pomeranian Medical University, 70-210 Szczecin, Poland; iwrot@wp.pl (I.R.); aleksandra.ryl@pum.edu.pl (A.R.); ptak.magda@gmail.com (M.P.); natalia.tomska@o2.pl (N.T.); aleksandra.szylinska@gmail.com (A.S.); 2Department of Anesthesiology, Intensive Therapy and Acute Intoxications, Pomeranian Medical University, 70-111 Szczecin, Poland; katarzyna.kotfis@pum.edu.pl; 3Department of Obstetrics and Pathology of Pregnancy, Pomeranian Medical University in Szczecin, 71-210 Szczecin, Poland; olimpiasipak-szmigiel@wp.pl

**Keywords:** depressive symptoms, magnesium, metabolic disorders, depressiveness

## Abstract

Background: changes in the concentration of magnesium influence numerous processes in the body, such as hormone and lipid metabolism, nerve conduction, a number of biochemical pathways in the brain, and metabolic cycles. As a result, changes in magnesium concentration may contribute to the emergence of such pathologies as depressive and metabolic disorders, including hypertension, diabetes, and dyslipidemia. Methods: blood samples were taken from 342 men whose mean age was 61.66 ± 6.38 years. The concentrations of magnesium, lipid parameters, and glucose were determined using the spectrophotometric method. Anthropometric measurements were performed to determine each participant’s body mass index (BMI). Additionally, all participants completed two questionnaires: the Beck Depression Inventory and the author’s questionnaire. Results: abnormal levels of magnesium were found in 78 people. The analysis showed that these subjects more often suffered from metabolic disorders such as diabetes mellitus (*p* < 0.001), hypertension (*p* < 0.001), and depressive symptoms (*p* = 0.002) than participants with normal magnesium levels. Conclusion: our research showed that there is a relationship between abnormal levels of magnesium and the presence of self-reported conditions, such as diabetes, hypertension, and depressive symptoms among aging men. These findings may contribute to the improvement of the diagnosis and treatment of patients with these conditions.

## 1. Introduction

Depressiveness is the occurrence of single symptoms, such as deterioration in mood; decreased energy and motivation; lowered quality of sleep; decreased libido, appetite, and self-esteem; anxiety without a specific cause; and cognitive and concentration disorders. These symptoms do not make up the full picture of the disease, the detailed criteria of which have been described in the 10th revision of the International Statistical Classification of Diseases and Related Health Problems (ICD-10) and in the Diagnostic and Statistical Manual of Mental Disorders, Fourth Edition (DSM-IV) published by the American Psychiatric Association (APA) [1]. Depressiveness is divided into mild, moderate, and severe, depending on the severity of a given symptom.

It is estimated that as many as 16.5% of people over 50 years of age suffer from depressive disorders in Western countries, and some other studies have shown that the risk of depressive episodes increases with age [2,3]. The reason why they are so common in this group may be due to changes in sufferers’ social level, such as a loss of one’s position in society, deterioration of the financial situation, and, often, a sense of loneliness and isolation [4]. At the physiological level, factors leading to a decline in mood include the degeneration of the nervous system and lower activity of noradrenergic and serotonergic neurotransmitters, causing negative changes in cognitive functions [5]. Damage to DNA and the cytoskeleton of the nerve cell is also associated with greater activity of the GABA-aminergic system and stimulation of NMDA (*N*-methyl-*D*-aspartate) and AMPA (α-amino-3-hydroxy-5-methyl-4-isoxazolepropionic acid) receptors, which results in greater Ca^2+^ transport. As a consequence, neurotoxic endonucleases and proteases are activated [6]. The level of bioelements in the blood plays an important role in the pathogenesis of depression at the physiological level. One of the most important, which influences a number of biochemical pathways, is magnesium. It acts as a blocker in the NMDA receptor, stopping the transport of Ca^2+^ ions and, at the same time, inhibiting the increased glutamatergic stimulation, which results in an environment characterized by high excitotoxicity. This environment, in turn, causes oxidative stress and, consequently, the degeneration and death of nerve cells [7]. The effect of magnesium on the receptors of other systems―namely the dopaminergic, noradrenergic, and serotonergic systems―has also been described in animal studies. In the case of the serotonergic system, the effect on the 5-HT1A and 5-HT2A receptors seems to be particularly important, as their activation plays a role in the inhibition of serotonin reuptake, which is also the mechanism of action of most antidepressants [8]. Meta-analyses examining the effect of low serum magnesium concentration on higher incidences of depression showed a potential correlation, but these conclusions were made with some caution [9,10] On the other hand, a meta-analysis of magnesium supplementation did not show any evidence of association with depressive disorders [11].

Unfortunately, the problem is further aggravated because, among people suffering from depressive symptoms, nutritional negligence often occurs, leading to a further decline in the level of magnesium in the blood. As a consequence, an increase in the severity of depression as well as other effects of this deficiency can be observed [12,13]. Some of the better-documented conditions are metabolic disorders, such as hypertension, diabetes, lipid metabolism disorders, and abdominal obesity. Magnesium seems to play an extremely important role in the regulation of blood pressure by inhibiting the renin-angiotensin-aldosterone system (RAAS) and angiotensin II in particular, which is responsible for narrowing blood vessels and increasing the heart rate [14]. It also stimulates the production of aldosterone, which is responsible for the reabsorption of sodium and fluids. Additionally, there was an inverse relationship between the levels of magnesium and aldosterone [15]. Another mechanism of the influence of the described bioelement on blood pressure is the reduction in calcium deposition in blood vessels, which improves the elasticity of blood vessels [16]. The mechanisms discussed were also confirmed in the meta-analyses performed on the impact of magnesium supplementation on the above arterial pressure. They showed an inverse correlation between magnesium intake and the occurrence of hypertension. Research on magnesium in sera also seems to support this relationship; however, the evidence here is somewhat more limited [17,18]. The role of magnesium in the pathogenesis of diabetes is associated with its participation in the phosphorylation of tyrosine kinase, which affects the activation of the insulin receptor [19]. Additionally, in the case of diabetes, meta-analyses seem to confirm the important role of the bioelement in question not only for glucose parameters but also for increasing insulin-sensitivity parameters in people at high risk of diabetes [20]. Other studies show a negative correlation between the level of magnesium and microalbuminuria [21]. There is also a positive effect of magnesium supplementation on the improvement of glucose uptake [22]. The effect of magnesium can be observed in the analysis of lipid metabolism. This element seems to have a beneficial effect on the reduction in cholesterol synthesis by inhibiting 3-hydroxy-3-methylglutaryl-coenzyme A (HMG-CoA) as well as on the reduction of low-density lipoprotein (LDL) by activating lecithin-cholesterol acyltransferase (LCAT) [23]. Additionally, magnesium contributes to a decrease in triglyceride levels and an increase in the level of high-density lipoprotein (HDL) through lipoprotein lipase (LPL).

Studies have shown that awareness of metabolic diseases such as hypertension is much lower in men than in women [24]. Additionally, there are unfavorable stereotypes that disregard and depress the problem of depressive disorders in men [25]. This is why it is so important to highlight the problem, especially in the aging group, which is most exposed to the above disorders. It is important to highlight this problem because these disorders are related to metabolic and hormonal changes, such as a decrease in testosterone concentration [26].

The aim of this study was to demonstrate the relationship between the level of magnesium and the presence of depressive symptoms and selected metabolic disorders in men over 50 years of age.

## 2. Materials and Methods

### 2.1. Characteristics of the Study Sample 

The study involved 342 male Caucasian participants between the ages of 50 and 77 (61.66 ± 6.38 years) who were recruited through primary health care centers in the city of Szczecin, Poland. Each participant received both oral and written information on the course of the study, had the opportunity to ask questions concerning his participation, was informed about the possibility of withdrawal at any stage, and signed a voluntary consent form to take part in the research. 

The subjects were divided into two groups according to the presence of depressive symptoms. The first group consisted of 249 participants showing no depressive disorders. The second group of 93 subjects included men with depressive symptoms.

### 2.2. Questionnaire Survey

Two questionnaires were used in the study: the Beck Depression Inventory-I (BDI-I) and the author’s questionnaire. The Beck Depression Inventory-I consists of 21 items concerning well-being, attitude towards one’s future, self-esteem, guilt, suicidal thoughts, level of irritation, interest in social life, appetite, weight loss, insomnia, fear for one’s own health, and sex drive. The patient independently chooses the sentence that best describes his or her emotional state during the last two weeks. Each item is rated on a 4-point scale from 0 (no symptoms) to 3 (severe symptoms). The points are then counted and compared with the norms provided for this questionnaire, whereby any score over nine points should be interpreted as positive for depression.

The author’s questionnaire concerned self-reported, basic sociodemographic data, the general health of the patient, including metabolic disorders, such as dyslipidemia, hypertension, and diabetes, as well as contraindications to participate in the study.

### 2.3. Determination of Magnesium by the Spectrophotometric Method 

Nine mL blood samples were drawn from the ulnar vein from all participants of the study. Tubes with a clot activator and a gel separator were used. The samples were centrifuged and poured into microtubes (1.5 mL). The material prepared in this way was stored at −80 °C.

To determine magnesium levels, inductively coupled plasma emission spectrometry (ICP-OES, ICAP 7400 Duo, Thermo Scientific Waltham, MA, USA) was used. First, 0.75 mL samples were thawed to room temperature and microwaved (MARS 5, CEM). Then, the material was transferred to polypropylene test tubes. Each tube was left for 30 min after a prior addition of 2 mL of 65% HNO3 (Suprapur, Merck). Then, 1 mL of 30% unstabilized H2O2 solution was added (Suprapur, Merck). The samples were heated in Teflon vessels for 35 min at 180 °C with microwaves. Then, the samples were cooled to room temperature and transferred to polypropylene tubes again, this time with a capacity of 15 mL. The next step was a 25-fold dilution. For this, 400 µL of the sample was taken, which was then enriched by adding the standard to obtain a final concentration of 0.5 mg/L of Ytrium 1 mL of 1% Triton (Triton X-100, Sigma Darmstadt, Germany). The final step was to dilute the sample to 10 mL with 0.075% nitric acid (Suprapur, Merck). The samples were then stored at a temperature of 8 °C. Preparation of the blank required the addition of 300 µL of nitric acid and its dilution as the test sample. Then, calibration standards for magnesium and phosphorus ICP (inductively coupled plasma) standard (AccuStandard, Inc., New Haven, CT, USA) were prepared with different concentrations of this element, similarly to the blank and test samples. Deionized water (Direct Q UV, Millipore, approx. 18.0 MΩ) was used. The magnesium concentration was read at a wavelength of 285.213 nm.

### 2.4. Determination of Lipids and Glucose by Spectrophotometric Method 

The following metabolic parameters were determined in serum by the spectrophotometric method using a standard procedure based on a ready-to-use reagent kit (Biolabo, Aqua-Med, Łódź): fasting plasma glucose (FPG), high-density lipoprotein (HDL), low-density lipoprotein (LDL), and triglycerides (TG).

### 2.5. Anthropometric and Blood Pressure Measurements 

All participants of the study had their weight and height measured in order to calculate the body mass index (BMI) according to the formula: BMI = body weight (kg)/height^2^ (m). The standards recommended by the World Health Organization (WHO) were used to interpret the result. Blood pressure was also measured in each patient. 

### 2.6. Statistical Analysis

All data were analyzed using licensed software Statistica 13 (StatSoft, Inc. Tulsa, OK, USA). The study sample was described in terms of the mean, standard deviation, median, as well as numbers and percentages. The normality of the distribution was assessed using the Shapiro–Wilk test. Analysis of differences between the two groups was performed using the Mann–Whitney U test. The chi-square test and the Yates correction were used to analyze qualitative data. The search for the optimal cut-off point for the level of magnesium was performed using the ROC (Receiver Operating Characteristic) curve. The evaluation of the relationship between the level of magnesium and selected parameters was performed with the use of multivariate logistic regression. The value of *p* < 0.05 was regarded as statistically significant.

## 3. Results

Data of 342 participants of the study were analyzed. Depressive symptoms were observed in 93 men, which constituted 27.19% of all respondents. The data collected from the interview and the results of laboratory tests are presented in Table 1.

The comparison of the mean results with regard to depressive symptoms showed statistically significant differences only for the levels of magnesium (*p* = 0.032). The results are presented in Table 2.

The levels of magnesium were analyzed with regard to depressive symptoms. The cut-off point for the level of magnesium was searched for. For this purpose, the ROC curve analysis was performed. The cut-off point was 19.83 mg/L. The ROC curve is presented in Figure 1.

The patients were divided according to the cut-off obtained from the ROC curve. Seventy-eight men (22.9%) had magnesium levels ≤19.83 mg/L and 263 men (77.1%) had magnesium levels >19.83 mg/L. Analysis of selected data with regard to the level of magnesium (Table 3) revealed that the number of patients with diabetes mellitus (*p* < 0.001), hypertension (*p* < 0.001), and depressive symptoms (*p* = 0.002) was higher among those with magnesium levels ≤19.83 mg/L. The level of magnesium >19.83 was more often associated with significantly higher LDL cholesterol levels (*p* < 0.001).

Table 4 shows the relationship between the level of magnesium ≤19.83 and selected parameters and factors. Multivariate regression analysis adjusted for age, smoking, education, marital status, and employment status showed a significant increase in the incidence of diabetes (OR = 4.328, *p* < 0.001), hypertension (OR = 2.613, *p* = 0.002), and depressive symptoms (OR = 2.102, *p* = 0.013) in people with magnesium levels ≤19.83.

## 4. Discussion

### 4.1. Magnesium and Depressive Disorders 

The study demonstrated a significant correlation between emotional disorders and decreased levels of magnesium in the blood serum. This phenomenon can be attributed to the mechanisms of magnesium influence on depressive disorders. The role of magnesium as a NMDA receptor blocker is most frequently described in the literature [27]. In the case of a deficiency of this element, the channels for the influx of calcium and sodium ions are opened [7]. According to Castilho et al., it increases glutamatergic neurotransmission, which, by stimulating the production of reactive oxygen species (ROS) and nitric oxide, contributes to the formation of a neurotoxic environment. This neurotoxic environment results in oxidative stress, edema, and cell death [28]. Many researchers, including Eby et al., confirm the role of the described mechanism in the development of emotional disorders [29]. Authors, such as Murek et al., emphasize another possible mechanism of the influence of magnesium on depression, namely, the role of this element in dysregulation of the hypothalamic–pituitary–adrenal (HPA) axis [30]. The animal study conducted by Cardoso et al. showed that the relationship between magnesium and the development of depression and depressive disorders may be associated with the influence of this bioelement on the dopaminergic, noradrenergic, and serotonergic systems through the 5-HT1A and 5-HT2A receptors [8].

However, not all studies indicated a significant correlation between the level of magnesium and depressive disorders and symptoms. Imada et al. observed higher magnesium levels among patients suffering from clinical depression [31]. The ambiguity of the available results suggests that a decrease in the level of magnesium in the blood serum is only perceived as a hypothetical cause of the exacerbation of depressive symptoms. In their work, Eby et al. suggested that the measurement of this element in tissues is much more precise and convincing for the clinician [32].

### 4.2. Magnesium and Metabolic Disorders 

Our research demonstrated a relationship between the level of magnesium and selected metabolic disorders, especially diabetes and hypertension. These results are consistent with findings of other authors dealing with the topic [33]. Mather et al., who analyzed 582 participants suffering from diabetes and 140 participants from the control group, found that those with diabetes had significantly lower levels of magnesium in the blood serum [34]. Similar conclusions were reached as early as in 1985 by Vanroellen et al., who observed hypomagnesemia in patients with non-insulin-dependent diabetes [35]. Djurhuus et al. examined both people suffering from insulin-dependent and non-insulin-dependent diabetes and noticed that both these groups had a low-magnesium diet [36]. In the case of Schmidt et al.’s research, only about 20% of the surveyed participants exceeded the recommended daily consumption of magnesium, i.e., 300 mg for women and 350 mg for men [37]. Mooren et al., on the other hand, found that supplementation with magnesium significantly increased insulin sensitivity in people who did not have any symptoms of metabolic syndrome [38]. This may prove the importance of supplementation with this bioelement for the prevention of metabolic dysfunctions. On the other hand, although Lima de Souz et al. confirmed the frequency of magnesium deficiency in people with metabolic syndrome; they neither observed a significant difference in insulin sensitivity nor an increase in the level of magnesium in the case of its supplementation with a dose of 400 mg for 12 weeks [39]. It is worth noting, however, that the study sample consisted of only 37 people, which may have been insufficient to detect such correlations.

The results showing the influence of magnesium on blood pressure have also been repeatedly reported by other authors. McCarron et al. observed a relationship between a decrease in blood pressure and supplementation with this bioelement [40]. In later years, these conclusions were positively verified by Rodríguez-Moran et al. and Guerrero-Romero et al. [41,42]. The results obtained by Villa-Bellosta and Louvet et al. suggest a significant role of magnesium deficiency in the formation of calcification in blood vessels, which is one of the risk factors for hypertension [43,44]. These findings were later substantiated by Hruby’s team [45]. According to Ichihara et al., supplementation with the bioelement in question influences the RAAS by reducing the activity of angiotensin II [46]. In 1997, Galan provided evidence for the effect of magnesium on the activation of the hypothalamic–pituitary–adrenal axis and the resulting stimulation of the autonomic nervous system [47]. However, the literature also includes studies that do not confirm the relationship between magnesium supplementation and positive changes in blood pressure [48].

The biggest limitation of our research was the ambiguity of the criteria for determining magnesium deficiency. Additionally, in a cross-sectional study it is impossible to assess a temporal relationship between exposure and outcome because both are assessed simultaneously. Another problem was a small study sample. We consider it necessary to conduct a study on a larger population.

## 5. Conclusions

Our research showed that there is a relationship between disturbed levels of magnesium and the presence of diabetes, hypertension, and depressive symptoms among aging men. These findings may contribute to the improvement of the diagnosis and treatment of patients with these conditions.

## Figures and Tables

**Figure 1 life-11-00196-f001:**
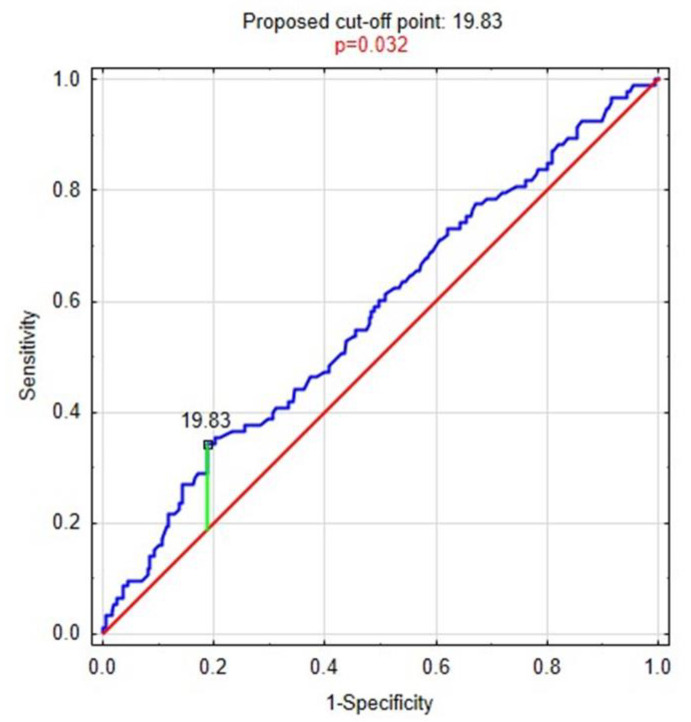
The ROC curve for magnesium levels.

**Table 1 life-11-00196-t001:** Characteristics of the study sample.

**Age (Years) Mean ± SD; Me**		61.66 ± 6.38; 62.00
Education (*n*, %)	Vocational	59 (17.25%)
	Primary	19 (5.56%)
	Secondary	140 (40.94%)
	Higher	119 (34.80%)
Smoking (*n*, %)		55 (16.08%)
Marital status (*n*, %)	Married	72 (21.05%)
	Unmarried	267 (78.07%)
BMI (*n*, %)	18.5–24.9	66 (19.30%)
	25.0–29.9	177 (51.75%)
	30.0–34.9	72 (21.05%)
	35.0–39.9	17 (4.97%)
	≥40.0	8 (2.34%)
BMI (kg/m2); mean ± SD; Me		28.41 ± 4.40; 27.77
Classification WHR (*n*, %)	≤1.0	154 (64.98%)
	>1.0	83 (35.02%)
WHR mean ± SD; Me		0.99 ± 0.07; 0.99
Hip circumference (cm) mean ± SD; Me		103.10 ± 7.83; 103
Abdomen circumference (cm) mean ± SD; Me		102.20 ± 12.08; 100
TAG (mg/dL); mean ± SD; Me		145.26 ± 82.92; 121.82
Cholesterol (mg/dL); mean ± SD; Me		216 ± 56.98; 211.69
HDL (mg/dL); mean ± SD; Me		48.33 ± 12.32; 46.45
LDL (mg/dL); mean ± SD; Me		140.69 ± 59.77; 133.48
Diabetes mellitus (*n*, %)		56 (16.37%)
Glucose (mg/dL); mean ± SD; Me		106.73 ± 38.59; 103.06
Hypertension (*n*, %)		190 (55.56%)
RR systolic; mean ± SD; Me		135.18 ± 19.71; 135
RR diastolic; mean ± SD; Me		88.47 ± 56.17; 80
Insulin (µLU/mL); mean ± SD; Me		14.84 ± 7.66; 13.45
HOMA; mean ± SD; Me		3.94 ± 2.22; 3.44
Serum Mg (mg/L); mean ± SD; Me		21.22 ± 2.52; 21.05

Abbreviations: WHR—waist-hip ratio; TAG—triacylglyceride; HDL—high-density lipoprotein; LDL—low-density lipoprotein; BMI—body mass index; RR—respiratory rate; HOMA—homeostatic model assessment; Mg—magnesium; SD—standard deviation; Me—median.

**Table 2 life-11-00196-t002:** Selected parameters in patients with and without depressive symptoms.

	Without Depressiveness (*n* = 249)	With Depressiveness (*n* = 93)	*p*
BMI (kg/m2); mean ± SD; Me	28.17 ± 4.1; 27.73	29.02 ± 5.09; 28.58	0.169
Abdomen (cm) mean ± SD; Me	101.82 ± 11.25; 100	103.19 ± 14.01; 101	0.604
TAG (mg/dL); mean ± SD; Me	142.66 ± 80.98; 121.82	152.18 ± 87.95; 121.83	0.343
Cholesterol (mg/dL); mean ± SD; Me	215.11 ± 55.25; 213.96	218.48 ± 61.62; 209.04	0.837
HDL (mg/dL); mean ± SD; Me	48.37 ± 12.62; 46.53	48.24 ± 11.55; 46.18	0.94
LDL (mg/dL); mean ± SD; Me	141.02 ± 59.60; 133.98	139.81 ± 60.53; 132.79	0.888
Diabetes mellitus (*n*, %)	37 (14.98%)	19 (20.43%)	0.227
Hypertension (*n*, %)	133 (53.85%)	57 (61.29%)	0.219
Serum Mg (mg/L); mean ± SD; Me	21.42 ± 2.49; 21.2	20.69 ± 2.56; 20.8	0.032

Abbreviations: TAG—triacylglyceride; HDL—high-density lipoprotein; LDL—low-density lipoprotein; BMI—body mass index; Mg—magnesium; SD—standard deviation; Me—median, *p*—statistical significence.

**Table 3 life-11-00196-t003:** Comparison of selected parameters and factors in both studied groups of men divided according to the results of the ROC curve.

	Mg ≤ 19.83 (*n* = 78)	Mg > 19.83 (*n* = 263)	*p*
BMI (kg/m2), mean ± SD; Me	28.89 ± 4.36; 28.31	28.27 ± 4.42; 27.73	0.244
Abdomen (cm), mean ± SD; Me	104.51 ± 11.39; 101.50	101.51 ± 12.22; 100.00	0.056
TAG (mg/dL), mean ± SD; Me	148.97 ± 82.89; 120.67	144.14 ± 83.05; 121.82	0.716
Cholesterol (mg/dL), mean ± SD; Me	193.27 ± 50.49; 190.96	222.85 ± 57.13; 221.65	<0.001
HDL (mg/dL), mean ± SD; Me	47.59 ± 12.20; 45.16	48.56 ± 12.37; 46.59	0.801
LDL (mg/dL), mean ± SD; Me	117.37 ± 49.51; 117.45	147.69 ± 60.88; 143.73	<0.001
Diabetes mellitus (*n*, %)	28 (35.90%)	28 (10.81%)	<0.001
Hypertension (*n*, %)	57 (73.08%)	130 (50.19%)	<0.001
Depressiveness (*n*, %)	32 (41.03%)	61 (23.37%)	0.002

Abbreviations: TAG—triacylglyceride; HDL—high-density lipoprotein; LDL—low-density lipoprotein; BMI—body mass index; Mg—magnesium; SD—standard deviation; Me—median, *p*—statistical significence.

**Table 4 life-11-00196-t004:** Multivariate analysis of selected data in people with magnesium levels ≤19.83.

	Unadjusted				Adjusted by Age, BMI, WHR, Smoking, Education, Marital Status, Professional Activity			
	*p*	OR	−95%	95%	*p*	OR	−95%	95%
BMI (kg/m2)	0.288	1.031	0.975	1.090	0.101	1.052	0.990	1.117
Abdomen (cm)	0.058	1.020	0.999	1.041	0.021	1.026	1.004	1.049
TAG (mg/dL)	0.652	1.001	0.998	1.004	0.472	1.001	0.998	1.004
Cholesterol (mg/dL)	<0.001	0.990	0.985	0.995	<0.001	0.989	0.983	0.994
HDL (mg/dL)	0.545	0.994	0.973	1.015	0.290	0.988	0.966	1.010
LDL (mg/dL)	<0.001	0.989	0.984	0.994	<0.001	0.988	0.983	0.994
Diabetes mellitus	<0.001	4.620	2.519	8.473	<0.001	4.614	2.451	8.686
Hypertension	<0.001	2.693	1.544	4.699	<0.001	2.833	1.584	5.069
Depressiveness	0.003	2.281	1.336	3.893	0.009	2.135	1.210	3.769

Abbreviations: WHR—waist-hip ratio; TAG—triacylglyceride; HDL—high-density lipoprotein; LDL—low-density lipoprotein; BMI—body mass index; Mg—magnesium; OR—odds ratio.

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
