# Peer review of "The Relationship between the Concentration of Magnesium and the Presence of Depressive Symptoms and Selected Metabolic Disorders among Men over 50 Years of Age"

_life, 2021, doi:10.3390/life11030196_

Round 1
Reviewer 1 Report
This is a cross-sectional study to correlate magnesium concentration of the blood with depressive symptom and some metabolic diseases in a few hundreds of adults older than 50 years of age.
Title and abstract
It is not appropriate to use the word “occurrence” of depressive symptoms, since this study did not observe any event to occur. It is better to use the word of “presence” instead of “occurrence” to show the cross-sectional nature of the findings. The metabolic diseases were self-reported. It is better to use the term of “self-reported” diseases in the abstract.
Introduction
It focused too much on mechanisms linking magnesium and depression and metabolic diseases. There were many epidemiological studies on the relationship between magnesium and hypertension, diabetes, and depression, including not only the blood concentration studies but also the dietary magnesium ones. Authors should make the current “status” of this topic clear in the introduction. Findings of meta-analysis should be mentioned to give a bigger picture, not just talking about a few studies.
Results
It is not clear what BMI groups 0-4 and WHR groups 0-1 mean in Table 1. The definition should be provided for table 1. In all tables, too many numbers after decimal points are given. The content in the abstract does not reflect what are included in the results. In the abstract it focused on the association between blood magnesium and depression and metabolic disorders, expressed by odds ratio. Yet in the results, comparisons were between two magnesium groups with respect to continuous clinical variables.
Discussion
It is important to discuss the limitation of cross-sectional study and not to imply the causal relationship between magnesium nutrition and depression.

Author Response
Response to Reviewer 1
Thank you so much for your review and suggestions to help us improve our manuscript.
Point 1: It is not appropriate to use the word “occurrence” of depressive symptoms, since this study did not observe any event to occur. It is better to use the word of “presence” instead of “occurrence” to show the cross-sectional nature of the findings. The metabolic diseases were self-reported. It is better to use the term of “self-reported” diseases in the abstract.
Response 1: Thank you very much for your comments. We replaced the word "occurrence" with "presence" in the text and title, and added the wording "self-reported" for metabolic diseases.
Point 2: It focused too much on mechanisms linking magnesium and depression and metabolic diseases. There were many epidemiological studies on the relationship between magnesium and hypertension, diabetes, and depression, including not only the blood concentration studies but also the dietary magnesium ones. Authors should make the current “status” of this topic clear in the introduction. Findings of meta-analysis should be mentioned to give a bigger picture, not just talking about a few studies.
Response 2: Thank you very much for your comments. In the introduction, we have added a meta-analysis on the relationship between magnesium supplementation and blood concentration of magnesium and diabetes, hypertension and depression.
Point 3: It is not clear what BMI groups 0-4 and WHR groups 0-1 mean in Table 1. The definition should be provided for table 1. In all tables, too many numbers after decimal points are given. The content in the abstract does not reflect what are included in the results. In the abstract it focused on the association between blood magnesium and depression and metabolic disorders, expressed by odds ratio. Yet in the results, comparisons were between two magnesium groups with respect to continuous clinical variables.
Response 3: Thank you very much for your comments. We defined the BMI and WHR designations in Table 1 and reduced the number of decimals to two. We also improved the consistency of the results between the abstract and the "results" section.
Point 4: It is important to discuss the limitation of cross-sectional study and not to imply the causal relationship between magnesium nutrition and depression.
Response 4: Thank you very much for your comments. We have added a description of the limitations of our study in the last paragraph of the discussion. And we also removed the last sentence that could have been imply the causal relationship between magnesium nutrition and depression.

Reviewer 2 Report
This manuscript describes a positive correlation between low plasma magnesium levels and the increased incidence of depressive and metabolic disorders in Polish men over 50 years of age. The manuscript focuses on a clinically relevant issue and is well written.
Author Response
Response to Reviewer
Thank you very much for your review of our manuscript and your positive feedback.
Sincerely,
Adrian Wiatrak

Reviewer 3 Report
This study investigated the relationship between the concentration of magnesium and the depressive symptoms and metabolic disorders among the limitation by men over 50 years of age. The results obtained are interesting and give an important suggestion on the important role of magnesium related to those background. However, several corrections are necessary on publishing the article.
First, in line 20, it is necessary for ‘concentractions’ to revise ‘concentration’.
Second, in line 121 and 179, it is necessary for ‘9’ and ‘78’ to correct ‘Nine’ and ‘Seventy-eight’, respectively. Because it is better not to state the number when a Roman numeral comes to at the beginning of a paragraph.
Author Response
Response to Reviewer
Thank you so much for your review and suggestions to help us improve our manuscript.
Point 1: In line 20, it is necessary for ‘concentractions’ to revise ‘concentration’.
Response 1: Thank you very much for your comment. Of course, the word ‘concentractions’ was corrected to ‘concentration’.
Point 2: In line 121 and 179, it is necessary for ‘9’ and ‘78’ to correct ‘Nine’ and ‘Seventy-eight’, respectively. Because it is better not to state the number when a Roman numeral comes to at the beginning of a paragraph.
Response 2: Thank you very much for your comment. All numbers have been replaced with a word.
